# The Structure and Function of Biomaterial Endolysin EFm1 from *E. faecalis* Phage

**DOI:** 10.3390/ma15144879

**Published:** 2022-07-13

**Authors:** Xuerong Zhou, Xiaotao Zeng, Li Wang, Yanhui Zheng, Guixiang Zhang, Wei Cheng

**Affiliations:** 1State Key Laboratory of Biotherapy, Sichuan University, Chengdu 610041, China; xuerong_zhou@163.com (X.Z.); 900592@163.com (L.W.); 2Division of Respiratory and Critical Care Medicine, Respiratory Infection and Intervention Laboratory of Frontiers Science Center for Disease-Related Molecular Network, State Key Laboratory of Biotherapy, West China Hospital of Sichuan University, Chengdu 610041, China; zengxt@scu.edu.cn (X.Z.); yahuzheng@outlook.com (Y.Z.); 3Laboratory of Bariatric and Metabolic Surgery, Department of Gastrointestinal Surgery, West China Hospital, Sichuan University, No. 37, Chengdu 610041, China; zhangguixiang@wchscu.cn

**Keywords:** *E. faecalis*, endolysin EFm1, crystal structure, lytic activity, CRISPR interference

## Abstract

The endolysin EFm1 from the *E. faecalis* 002 (002) phage IME-EF1 efficiently lyses *E. faecalis*, a gram-positive bacterium that severely threatens human health. Here, the structure and lytic activity of EFm1 toward *E. faecalis* were further investigated. Lytic activity shows that EFm1 specifically lyses 002 and 22 other clinically isolated *E. faecalis*, but not *E. faecalis* 945. Therefore, EFm1 may be an alternative biomaterial to prevent and treat diseases caused by *E. faecalis*. A structural analysis showed that EFm1_D166Q_ is a tetramer consisting of one full-length unit with additional C-terminal domains (CTDs), while EFm1_166–237 aa_ is an octamer in an asymmetric unit. Several crucial domains and novel residues affecting the lytic activity of EFm1 were identified, including calcium-binding sites (D20, D22 and D31), a putative classic amidohydrolase catalytic triad (C29, H90 and D108), a tetramerization site (M168 and M227), putative ion channel sites (IGGK, 186–198 aa), and other residues (R208 and Y209). Furthermore, EFm1 exhibited no significant activity when expressed alone in vivo, and IME-EF1 lytic activity decreased when *efm1* was knocked down. These findings provide valuable insights into the molecule mechanism of a potential functional biomaterial for the treatment of the disease caused by the opportunistic pathogen *E. faecalis*.

## 1. Introduction

The emergence of antimicrobial resistance in microorganisms is a global concern and poses a serious threat to public health [1,2]. *E. faecalis*, a gram-positive bacterium that is commonly present in humans and animals, is an opportunistic pathogen that causes community and hospital-acquired infections and can be life-threatening to immunocompromised patients [3]. The cell wall of *E. faecalis* is mainly composed of peptidoglycan, teichoic acid and lipoteichoic acid. Lipoteichoic acid has a negative charge and is covalently bound to the cell membrane; its chain extends into the cell wall [4]. It is extremely difficult to use routine antibiotics to eliminate *E. faecalis* strains with high levels of antibiotic resistance, especially vancomycin-resistant *E. faecalis* (VREF) [5]. By modifying the binding site, D-alanyl-D-alanine (terminal residues of stems) was converted to D-alanyl-D-lactic acid in vancomycin-resistant *E. faecalis*, resulting in a 1000-fold decrease in its binding affinity for vancomycin [6]. Mobile genetic elements, including plasmids and transposons, play a critical role in the spread and persistence of drug resistance in *E. faecalis* [7]. Therefore, the search for new drugs to treat infections caused by this opportunistic pathogen is urgent.

Phage therapy is regarded as a feasible alternative treatment option when antibiotics are ineffective against bacterial infections, and endolysin (lysin) therapy is an important part of this strategy [8]. Phage-encoded endolysins are highly evolved peptidoglycan hydrolases that lyse the host cell at the terminal stage of the phage lytic lifecycle to ensure the release of phage progeny [9]. Endolysins are considered alternative therapeutic agents for use against organisms with drug tolerance [10]. Endolysins not only show high lytic potential against numerous bacterial species, but also provide an approach to eliminate biofilms [11]. There are several advantages associated with the use of endolysin to treat infections caused by this opportunistic pathogen, such as high efficiency and specificity, low resistance and few neutralizing antibodies [12]. Normally, lyase cannot remain in a stable state at room temperature. However, after bonding to a surface or inside specific materials, its stability is improved, making it a functional biomaterial [13].

An *E. faecalis* 002 (002) bacteriophage (phage) named IME-EF1 was isolated from hospital sewage. A whole-genome sequence analysis showed that this phage belonged to the *Siphoviridae* family; it was found to be highly homologous with *E. faecalis* phage SAP6 and *E. faecalis* phage BC-611 [14]. EFm1, a lysin of IME-EF1, is similar to the lysin from SAP6 and BC-611 in the predicted amino acid sequence and conserved domain [14]. These three lysins all have a conserved CHAP (cysteine-histidine-dependent amido-hydrolase/peptidase) domain in the N terminal, while EFm1 possesses a much broader bactericidal spectrum including two VREF strains (*E. faecalis* V583). Similar to LysEF-P10 [14], EFm1 decreased fatality rates to 20% within 30 min after intraperitoneal injections were given to mice with 002 [15]. Zhou et al. reported that EFm1 exhibited a unique architecture in which one full-length forms a tetramer with three additional C-terminal cell-wall binding domains (CBD). *EFm1* contains an internal ribosomal binding site upstream of a putative alternative start codon, which is responsible for the additional CBDs that are important for its activity [16].

Therefore, EFm1 is expected to be a novel drug to treat diseases caused by the ectopic parasitism of *E. faecalis*, such as sepsis and urinary tract infections. However, the lytic spectrum of EFm1 to *E. faecalis*, the crucial domains and residues affecting its lytic activity and its role in phage IME-EF1 infection need to be further explored. In this study, the *E. faecalis* 945 strain resisted to Efm1 was obtained by a deep broadening of the cleavage spectrum of EFm1. Additionally, the lytic mechanism of EFm1 was identified by determining the structures of EFm1_D166Q_ and EFm1_166–237 aa_, analyzing their lytic activity and verifying the activity of *efm1* in 002 and IME-EF1.

## 2. Materials and Methods

### 2.1. Phage and Strains

The *E. faecalis* phage IME-EF1 (GenBank accession number: NC_041959.1) and cultures of the *E. faecalis* strains 002, 19, 255, 281, 284, 285, 410, 436, 462, 620, 630, 673, 697, 850, 863, 945, 1550, 1590, 1591, 1592, 1593 and 1763 were acquired from Tong Yi Gang (Anhui Medical University). The *E. faecalis* strains V583 and ATCC29212 were obtained from Huang Xiao Jing (Fujian Medical University) and Cheng Li (West China School/Hospital of Stomatology Sichuan University), respectively. *E. faecalis* cultures were inoculated at 37 °C in brain heart infusion (BHI) medium (USA, BD Difco; for 1 L total: 7.7 g of calf brain, infusion from 200 g; 9.89 g of beef heart, infusion from 250 g; 10.0 g of proteose peptone; 2 g of dextrose; 5.0 g of sodium chloride; and 2.5 g of disodium phosphate; final pH 7.4 ± 0.2). *E. coli* BL21(DE3) and *B. subtilis* CU1050 [17] were preserved in our laboratory and cultured in Luria-Bertani (LB) medium (for 1 L total: 10 g of NaCl (China, Dragon), 10 g of tryptone (USA, OXOID), 5 g of yeast extract (USA, OXOID)).

### 2.2. Plasmid Construction

The codon-optimized *efm1* (GenBank accession number: YP_009603970) from IME-EF was synthetized by GENEWIZ and cloned into the vector pGEX-6P-1 (addgene plasmid: 113500) to produce the plasmid pGEX-EFm1. All plasmids expressing EFm1 mutants were constructed by the Quik Change method with the corresponding primers using pGEX-EFm1 as a template [17]. The primers used in this study are listed in Table 1. The sequences of all constructs used in this study were confirmed by DNA sequencing. The operations performed were as follows:

The XtalPred [18] and SERp [19] servers were used to predict disordered loops and identify high-entropy surface patches of EFm1, since the wild-type EFm1 crystal could not be harvested in our experiment. Based on the results, the D166Q mutant of EFm1 was used for crystallization; the plasmid pGEX-EFm1_D166Q_ was constructed using the primers EFm1_D166Q_-F/R, and the purified protein was used to solve the crystal structure of EFm1_D166Q_.

The *efm1* fragment was amplified with the primers pDL278-EFm1-F/R from the plasmid pGEF-6P-1 and cloned into the plasmid pDL278 (addgene plasmid: 46882) to produce pDL278-*efm1*. CRISPR interference (CRISPRi) was used to interfere with the expression of *efm1* in vivo with the plasmid pDL278-dCas9-sgRNA. The dCas9 fragment was amplified with the primers dCas9-F/R from the plasmid pRH2502 (addgene plasmid: 84379) and cloned into the plasmid pDL278 to produce pDL278-dCas9. Subsequently, the plasmid pDL278-dCas9 was linearized by amplification with the primers pDL278-F/R. The sgRNA scaffold containing *efm1*-targeting sgRNA was amplified with the primers sgRNA-F/R from the plasmid pJMP3-yorR [17], which was previously constructed by our laboratory and cloned into linearized pDL278-dCas9 to produce pDL278-dCas9-sgRNA.

### 2.3. Protein Expression and Purification

The expression and purification of EFm1 were performed as previously described with minor modifications [17,20]. The isoelectric point (pI) of EFm1 predicted by EditSep software was 6.75. A single colony of *E. coli* BL21 (DE3) containing a plasmid encoding wild-type EFm1 or a variant was inoculated into 10 mL of LB medium with 100 µg/mL ampicillin and grown for 12–15 h at 37 °C. Then, the culture was used to inoculated 1 L of LB medium with 100 µg/mL ampicillin. Cultures were grown at 37 °C until the absorbance at 600 nm (OD_600 nm_) reached 0.6–0.8. Cultures were grown at 16 °C for 18 h, and isopropyl β-D-thiogalactoside (IPTG) was added at a final concentration of 3 mM to induce protein expression. Harvested cell pellets were resuspended and lysed by sonication in buffer A (25 mM Tris-HCl, 200 mM NaCl; pH 8.0). Then, the cell lysate containing the N-terminal GST-EFm1 protein was applied to glutathione-coated Sepharose resin (GE Healthcare) and washed in buffer A. Subsequently, the protein was assessed by anion exchange chromatography (Source Q; GE Healthcare) with buffer B (25 mM Tris-HCl; pH 8.0) and buffer C (25 mM Tris-HCl, 2 M NaCl; pH 8.0) and size-exclusion chromatography (columns from GE Healthcare) with buffer A for further purification. The peak fractions were collected and assessed by sodium dodecyl sulfate—polyacrylamide gel electrophoresis (SDS-PAGE). The clear fractions were pooled and flash frozen in liquid nitrogen for crystallization. Selenomethionine (Se-Met)-substituted proteins were expressed in M9 medium (5 × M9, for 1 L total: 44.3 g of Na_2_HPO_4_ (China, Dragon), 19.5 g of KH_2_PO_4_ (China, Dragon), 3.25 g of NaCl (China, Dragon), 6.5 g of NH_4_Cl (China, Dragon)).

### 2.4. Crystallization and Structure Determination

Crystals of EFm1_D166Q_ (15 mg/mL) and the C-terminal domain EFm1_166–237 aa_ (10 mg/mL) were grown at 16 °C by the sitting drop method, with one precipitant consisting of 0.1 M HEPES (pH 7.0–7.2), 0.2 M NaCl and 18–22% poly (ethylene glycol) 6000 and another consisting of 20% PEG 3350 and 0.2 M KCl. The Se-Met proteins of EFm1_D166Q_ (10 mg/mL) were crystalized under the same conditions. All of the crystals grew to full size in a week and were flash frozen in liquid nitrogen with an additional 10% glycerol as a cryo-protectant before X-ray diffraction. Crystallographic phases were determined by single-wavelength anomalous diffraction methods from data collected on crystals of Se-Met EFm1_D166Q_ and EFm1_166–237 aa_. All datasets were obtained at beamlines BL17U1 and BL19U1 at the Shanghai Synchrotron Radiation Facility (SSRF). The collected frames were processed and scaled using the XDS data processing package [21] and HKL2000 [22]. The Se-Met sites and density modification were identified by the HySS tool in phenix package and Resolve, respectively. An initial model was built and refined by Resolve in phenix.autobuild and phenix.refinement, respectively. The final model was built in COOT [23] and refined in Phenix2 [24].

### 2.5. Lytic Activity Experiment

The lytic activities of wild-type EFm1 and different variants were measured through plate lysis assays and the bacterial growth curve. In the plate lysis assay, wild-type EFm1 and different mutants (2 × 10^−7^ mol/L, 2 μL) were dropped onto a BHI semisolid culture medium containing *E. faecalis* 002 in the logarithmic phase, and the plates were observed after 12 h. In the bacterial growth curve assays, a starter culture was prepared using a colony from a freshly inoculated BHI agar plate to inoculate 5 mL of BHI medium. After overnight cultivation, the bacteria were inoculated into 10 mL of BHI medium at a ratio of 1:100 and incubated at 37 °C with shaking (220 rpm/min) until the OD_600 nm_ reached 0.1. The bacterial culture was centrifuged at 8000× *g* for 7 min at room temperature and resuspended in an equal volume of BHI medium. Buffer A was used as a negative control, and 200 µL of 002 was mixed with 2 µL of 2 × 10^−7^ mol/L wild-type EFm1 (positive control) or different mutants. OD_600 nm_ measurements were performed using a BioTek Cytation 5 Cell Imaging Multi-Mode Reader in a 96-well plate every 15 min for 12 h.

To determine the influence of metal ions (Ca^2+^, Cu^2+^, Fe^2+^, Ni^2+^, Mg^2+^ and Zn^2+^) and ethylene diamine tetraacetic acid (EDTA)—a chelating agent combined with divalent metal ions—on the lytic activity of EFm1, 200 µL of *E. faecalis* 002 was mixed with 1 mM (final concentration) metal ion or EDTA and 2 µL of 2 × 10^−7^ mol/L EFm1, and the OD_600 nm_ was measured every 15 min for 12 h. All the growth curves were repeated three times.

### 2.6. CRISPR Interference Experiments

pDL278-*efm1* was transformed into 002 by electrotransformation, and positive clones were screened from a BHI plate supplemented with 500 μg/mL spectinomycin. The fresh clones were cultured in BHI medium until the logarithmic growth phase (OD_600 nm_ = 0.1). The bacteria were harvested by centrifugation at 8000× *g* for 7 min and resuspended in equal volumes. Bacteria containing pDL278-sgRNA-dCas9 plasmid were infected with phages at a multiplicity of infection (MOI) of 0.1. The OD_600 nm_ was measured by a BioTek Cytation 5 Cell Imaging Multi-Mode Reader in a 96-well plate every 15 min for 5 h.

### 2.7. Statistical Analysis and Figure Preparation

The growth curve data are represented as means ± s.e.m. All figures were produced with PyMOL [25] and GraphPad Prism 8 [26].

## 3. Results

### 3.1. Lytic Activity of EFm1

EFm1 was expressed in *E. coli* BL21(DE3) from the pGEX-6P-1 plasmid and then purified. The SDS-PAGE results showed two bands which were further identified approximately 27 kDa and 8 kDa by mass spectrometric analyses (Figure 1A).

The results of the growth curve analysis showed that EFm1 could significantly inhibit the growth of *E. faecalis* 002, ATCC29212, and V583 until 9 h, in contrast to *E. coli* BL21(DE3) and *B. subtilis* CU1050 (Figure 1B–D). There was a slight lag in the growth of *E. faecalis* 002 compared with V583 and ATCC29212 (Figure 1B). These results indicated that EFm1 can specifically lyse *E. faecalis* 002, ATCC29212, and V583 but not *E. coli* BL21(DE3) and *B. subtilis* CU1050.

To further analyze and compare the lytic spectrum of wild-type EFm1 and EFm1_D166Q_, 23 clinically isolated strains of *E. faecalis* (002, 19, 255, 281, 284, 285, 410, 436, 462, 620, 630, 673, 697, 850, 863, 945, 1550, 1590, 1591, 1592, 1593 and 1763) were used for the plate lysis assay. The results showed that the lytic spectrum of EFm1_D166Q_ was the same as that of wild-type EFm1, and both of them can lyse 22 tested *E. faecalis* clinical strains, except for *E. faecalis* 945.

### 3.2. Structures of EFm1_D166Q_ and EFm1_166–237 aa_

The structures of EFm1_D166Q_ and EFm1_166–237 aa_ were determined to elucidate the molecular mechanism; the structure of EFm1_D166Q_ was successfully determined at 2.2 Å resolution in the P12_1_1 space group by X-ray crystallography. The structure of C-terminal EFm1 was refined at 2.0 Å in the I4_1_22 space group. EFm1_D166Q_ contains an NTD (1–145 aa), a CTD (165–237 aa) and a linker (145–165 aa) (Figure 2A). A CHAP domain is located in the NTD and the CTD forms a tetramer. The linker connects the NTD and CTD.

Similar to the lysin PlyC from *streptococcal* specific phage, of which CTDs are arranged in a planar octameric ring [27], the CTDs of EFm1_D166Q_ also are arranged in a planar tetrameric ring consisting of a CTD of full-length EFm1_D166Q_ and three CTDs (Figure 2B). The final refined structure of EFm1_D166Q_ possessed good stereochemistry except for residues 145–165, where the electron density was missing due to its high flexibility. The CHAP domain of EFm1_D166Q_ is composed of three α-helices, two 3_10_-helices and six β-strands (Figure 2B). Two α-helices are interconnected by a long loop (L1, 14–29 aa). Together, this loop and its neighbor loop (L2, 108–123 aa) form negative potential on the surface of the protein (Figure 2C). A hydrophobic groove is formed between the two loops and the two 3_10_-helices (Figure 2C). This arrangement might contribute to the catalytic activity responsible for the cleavage of peptidoglycan bonds.

The structure of EFm1_166–237 aa_ is the same as the C-terminal domain of the refined EFm1_D166Q_ (Figure 2D). The structure of EFm1_166–237 aa_ shows eight monomers in an asymmetric unit. However, extensive intermonomer interfaces occur only in every four monomers in the crystal, suggesting that in solution, the protein is tetrameric. Each monomer is composed of a four-stranded β-sheet capped on each side by a short α-helix, which is arranged in the characteristic α-β-α fold. The four β-strands together form the central β-sheet layer of EFm1-CTD. The “sandwich” conformation greatly stabilizes tetramerized EFm1. Notably, this tetramer structure maximizes the arrangement of internal hydrophilic residues and external hydrophobic residues with the formation of internal positively charged cavities (Figure 2E). Together, these results imply that the tetramerized EFm1-CTD might be crucial for phage anchoring on the peptidoglycan. The crystal data are shown in Table 2.

### 3.3. The Effect of the NTD, CTD and Linker on the Lytic Activity of EFm1

To assess the effect of the NTD, CTD and linker on the lytic activity of EFm1, we expressed the NTD, CTD and mutant lacking the linker (Figure 3A). SDS-PAGE of the mutant lacking the linker showed only a band slightly lower than that of wild-type EFm1. Therefore, we hypothesize that amino acids located at approximately residues 145–165 are responsible for generating the 8 kDa fragment. The growth curves and plate lysis assay showed that the NTD, CTD and mutant lacking the linker showed no lytic activity compared with wild-type EFm1 (Figure 3B,C). This indicated that only full-length EFm1 could lyse *E. faecalis*.

### 3.4. Binding Sites and Iron Channel of EFm1

To identify the region of the EFm1 tetramer that is involved in cell wall binding, we undertook a mutagenesis study, mutating residues in the concave region of EFm1. There was a prominent convex region in each monomer that was lined by residues K173, K176, R190 and K236, resulting in a positively charged patch surrounding the convex (Figure 4A). Thus, we hypothesize that the convex positively charged interface forms a site for binding the bacterial cell wall through ionic bonds. To verify the involvement of binding sites in this positively charged patch, we mutated residues K173, K176, R190 and K236 to glutamic acid. None of these mutations interfered with the formation of the EFm1 tetramer ring or the formation of the holoenzyme (Figure 4B). The results of growth curve and plate lysis assays indicated that the K173E, K176E and K236E mutants retained lytic activity, while the R190E mutant showed no lytic activity compared to wild-type EFm1. We hypothesize that R190 is the key residue mediating binding between EFm1 and *E. faecalis* (Figure 4C,D).

### 3.5. Lytic Activity of the CHAP Domain of EFm1

Similar to LysGH15 (PDB: 4OLK) and LysK (PDB: 4CSH), the CHAP domain of EFm1 contains a calcium ion which coordinates with the conserved residues D20, D22, W24, G26, and D31 and one water molecule (Figure 5A). The growth curves showed that there was no activity when EDTA was mixed with EFm1, while activity was restored after the addition of calcium, indicating that calcium is important during lysis (Figure 5B).

Eijsink et al. suggested that protease easily recruit metal ions to the active center in the calcium-rich environment of gram-positive cells; the stable states of protease are regulated by the number of calcium binding sites [28]. When all calcium binding sites are occupied, protease can fold to a stable, active conformation. Therefore, calcium usually serves as a switch to regulate the activity of proteases. As such, atoms coordinating with calcium could affect the activity of proteases by changing the spatial position of calcium, such as subtilisin or alpha-lytic protease [28]. Like LysGH15 [29], the mutants W24A and G26A of EFm1 remained active, while mutations at D20, D22 and D31 abolished activity (Figure 5C–E).

Besides calcium, there is a putative classic amidohydrolase catalytic triad consisting of residues C29, H90 and D108 in CHAP (Figure 6A). The putative catalytic triad is clearly next to the calcium, and a 2.8 Å distance was observed between H90 and D108. In addition, C29 and H90 are located in the cleft of the CHAP domain. The growth curves and plate lysis assay showed that there was no lytic activity among the C29A, H90A and D108A mutants (Figure 6B,D). Shin et al. reported that the nucleophilic residue (Ser or Cys) is most likely in the protonated form at physiological pH [30]. The nucleophile is maintained in the deprotonated state by the direct hydrogen-bonding interactions provided by the side-chain-NH and the side-chain-OH groups from acidic amino acids and basic amino acids, respectively. Then, nucleophilic attack by the nucleophile on the carbonyl carbon of the substrate results in the cleavage and formation of chemical bonds [30]. The three residues (C29, H90 and D108) forming the putative catalytic triad affect the lytic activity. It is possible that D108 bonds with H90 by hydrogen bonding, causing C29 deprotonation, which cleaves the chemical bonds in peptidoglycan. Therefore, calcium may also regulate protein activity by regulating C29 to place it in the correct spatial orientation, and coordinates with a putative classic amidohydrolase catalytic triad to contribute to the activity of EFm1.

### 3.6. Lytic Activity of the Tetramer of EFm1

We found that there was only one protein band for the mutant lacking the linker (Figure 3A), so we hypothesized that amino acids located at approximately residues 145–165 were responsible for generating the 8 kDa fragment. After analyzing the sequence of *efm1*, we hypothesized that M168 acted as an alternative start codon for in-frame translation. The mutant M168A peak appeared later than the wild-type EFm1 peak in the size-exclusion chromatography experiment (Figure 7B), and the 8 kDa band disappeared (Figure 7C). The above results indicated that M168 is responsible for the translation of the 8 kDa fragment.

Next, we tried to reveal the assembly mechanism of the 8 kDa fragment. By structural analysis, we found that M227 was bound to another monomer with a hydrophobic cage composed of F184, I191, L193, Y198, A202 and V237 (Figure 7A). In the size-exclusion chromatography experiment, the mutant M227K peak appeared later than the wild-type EFm1 peak (Figure 7B). By SDS-PAGE analysis, there was only one band at approximately 27 kDa (Figure 7C). The result for the M227A mutant was the same as that for wild-type EFm1 (data not shown), suggesting that recognition and interaction are mediated by hydrophobic interactions between the full-length unit and three additional fragments. The growth curves and plate lysis assay showed that the M168A and M227K mutants had no lytic activity, confirming that oligomeric tetramerization of EFm1 is essential for its lytic activity (Figure 7D,E).

### 3.7. Interaction of the Monomer and Ion Channel in the CTD

We know that there is a strong hydrophobic interaction between monomers and a large amount of hydrogen bonding. Structural analysis showed the presence of some hydrogen bonds between residues of the α-helix in one monomer and the β4-sheet in the adjacent monomer. Residue R208 forms hydrogen bonds with residues Y218 and A231 and residues Y209 and F169 of the adjacent protomer (Figure 8A). To assess the importance of these residues for tetramerization and activity, we constructed variants of EFm1. SDS-PAGE showed that R208A and Y209A did not interfere with the formation of the holoenzyme (Figure 8C), while the plate lysis assay showed that these variants were not active (Figure 8D).

There is a putative iron channel (I186-K189) in the CTD of EFm1 (Figure 8B) which was filled with positively charged residues. SDS-PAGE showed that this four-residue mutation did not interfere with the formation of the holoenzyme (Figure 8B), while the plate lysis assay showed that the mutants were not active (Figure 8D).

All the domains and residues identified in this study are summarized in Table 3.

### 3.8. Expression In Vivo and Knockdown of EFm1 in 002

The growth of 002 containing the plasmid pDL278-*efm1* was measured to investigate the lytic activity of EFm1 in vivo. From 3 h to 6 h, the growth of 002 containing the plasmid pDL278-*efm1* was slightly lower than that of the control (Figure 9A). After that, the growth dynamics of the two strains were similar. These results implied that EFm1-mediated lysis in the IME-EF may need holins for better activity.

In addition, we carried out a CRISPRi experiment to investigate whether silencing of *efm1* affected the activity of phage during infection (Figure 9B). The growth curves showed that from 2 h to 3 h, the growth of 002 containing pDL278-dCas9-sgRNA and infected with the phage IME-EF1 was higher than that of 002 containing pDL278 and infected with the phage IME-EF1, indicating that silencing *efm1* resulted in a decrease in the lytic activity of the phage IME-EF1 toward 002.

## 4. Discussion

The aims of this study were to investigate the lytic spectrum of EFm1 toward *E. faecalis*, identify the crucial domains and residues affecting the lytic activity of EFm1 and analyze the role of EFm1 in phage IME-EF1 infection. Based on a structural analysis, bacterial growth curves and plate lysis assays further identified some domains and novel residues that were important for the stability and lytic activity of EFm1, such as the NTD, CTD, linker, residues involved in host cell-binding sites, calcium-binding sites, the putative catalytic triad, 8 kDa assembly fragment, surface interaction sites and ion channel sites.

To analyze the lytic spectrum of EFm1, which specifically lyses *E. faecalis*, we tested its lytic ability against clinically isolated strains. EFm1 and EFm1_D166Q_ could lyse 22 clinical strains, except *E. faecalis* 945. We hypothesized that *E. faecalis* 945 may possess a unique capsule or biofilm or a hydrolase on the outer cell wall, helping it escape or resist the activity of EFm1, or absent tail fiber protein of phage which specifically degrades biofilm during the phage host interaction [31]. These implied that *E. faecalis* 945 might be the most suitable bacteria used to further investigate the resistance mechanisms of endolysins and the emergence of endolysin-resistant *E. faecalis* might bring new challenge for phage therapy.

In this research, the structure of EFm1_D166Q_ showed that there are three domains (a CTD, an NTD and a putative linker) in the holoenzyme. CHAP and calcium- or cell wall-binding domains are present at the N-terminus and C-terminus, respectively. Crystal data and biochemical experiments revealed that EFm1 contains four potential binding sites (R190) in the convex region of CTDs for cell wall components which may allow tighter, more stable interactions to occur between the holoenzyme and the cell wall, in contrast to most lysins containing only a single CBD [27,32]. However, we are not sure if all four binding sites simultaneously participate in coordinating holoenzyme binding to cell wall.

The results showed that there was no lytic activity when mutate residues coordinating calcium with were present in the side groups or formed putative classic amidohydrolase catalysts. In the highly conserved triad, the protonation of the nucleophilic residue by acid/base glutamate facilitates the first step of the catalytic reaction [33]. We speculate that lysis proceeds as follows: calcium regulates C29 to ensure the correct spatial orientation, and the -NH of H90 interacts with the -OH of D108, leading to the deprotonation of C29. Next, the highly flexible linker undergoes a conformational change to allow the N-terminus to reach the surface of the cell wall. Finally, the deprotonated C29 attacks the carbonyl carbon of the substrate, resulting in the cleavage of the chemical bonds of peptidoglycan.

M168, an alternative start codon for in-frame translation, is responsible for the generation of the other three fragments [16]. We have found that M227 inserting a hydrophobic pocket in the adjacent monomer is crucial to assemble the tetramer. Ortega et al. reported that NisB substrate recognition and interaction is mediated by hydrophobic interactions between NisB and NisA [34]. As such, multiple novel residues identified with different structural characteristics may affect the lytic activity of EFm1.

In this research, some of our results are consistent with those reported by Zhou et al. [16], such as the effects of residues C29, H90, M168, R190 or EDTA on the lytic activity of EFm1. Furthermore, some novel residues affecting the lytic activity of EFm1 were identified in this study. For example, although Zhou et al. reported that N110 was one of residues forming a putative catalytic triad, we found that residue D108 might also form the putative catalytic triad. Furthermore, we have proved that residues D20, D22 and D31 coordinated with calcium in its side groups or located between monomers R208 and Y209 were the key sites of activity. We also found that the hydrophobicity of M227 was responsible for the tetramer.

EFm1 exhibited no significant activity when expressed alone in vivo, and CRISPRi showed that 002 had a slight resistance to IME-EF when *efm1* was silenced. It is reported that holins make a hole in the bacterial cytoplasmic membrane to create a channel to help lysin delivery to the peptidoglycan and the release of phage progeny at the end of the lytic cycle [11,35,36]. It is possible that EFm1 cannot be effectively released in the absence of holins.

## 5. Conclusions

This research further expands the lytic spectrum of EFm1 to *E. faecalis*. It also identified the EFm1-resistant strain *E. faecalis* 945, obtained the structures of EFm1_D166Q_ and EFm1_166–237 aa_, identified novel crucial residues affecting the lytic activity of EFm1, and analyzed their role in phage infection. Notably, we could not clearly explain why the electron density in the linker loop was missing. Additionally, the cleavage mechanism of EFm1 needs further research. Further investigation of EFm1 will be important not only to elucidate the working mechanism of EFm1 but also to promote research into the preparation of functional biomaterials based thereon. This study provides a research basis for EFm1, which is becoming recognized as a new generation of antibacterial biomaterial and a viable choice for the treatment of diseases caused by the opportunistic pathogen *E. faecalis*.

## Figures and Tables

**Figure 1 materials-15-04879-f001:**
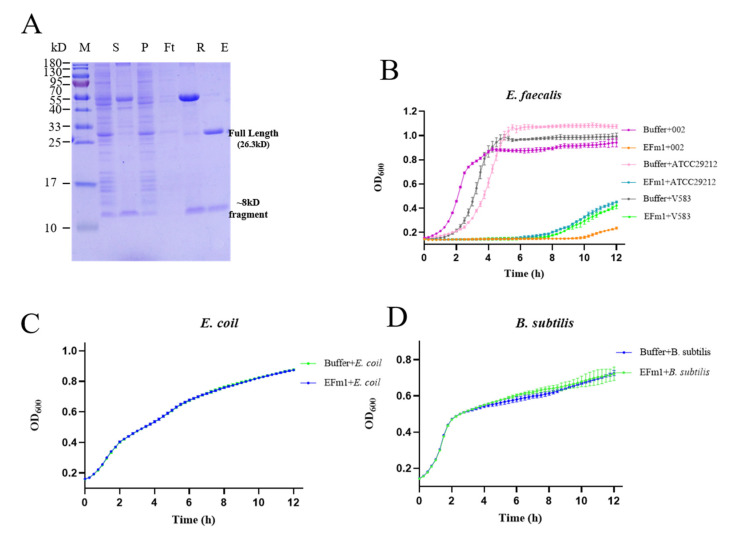
Expression and lytic activity of EFm1. (**A**) SDS-PAGE analysis of EFm1 expression. S, P, Ft, E and R represent supernatant, pellet, flow through, elution and resin, respectively; (**B**) The growth curves of 002, ATCC29212 and V583 after adding EFm1; the mean ± s.e.m is shown; (**C**) The growth curves of *E. coli* BL21(DE3) after adding EFm1; the mean ± s.e.m is shown; (**D**) The growth curves of *B. subtilis* CU1050 after adding; the mean ± s.e.m is shown.

**Figure 2 materials-15-04879-f002:**
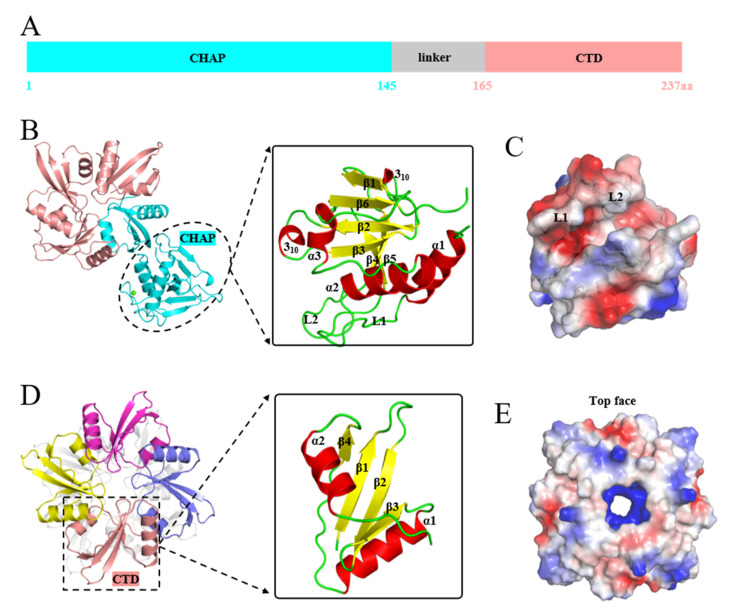
Three-dimensional structures of EFm1_D166Q_ and EFm1_166–237 aa_. (**A**) Domain organization of EFm1; (**B**) Structure of EFm1_D166Q_ including the NTD (1–165 aa), the CTD (165–237 aa) and a putative linker (145–165 aa); (**C**) Electron density maps of the CHAP domain; (**D**) Structure of EFm1_166–237 aa_; (**E**) Electron density map of EFm1_166–237 aa_.

**Figure 3 materials-15-04879-f003:**
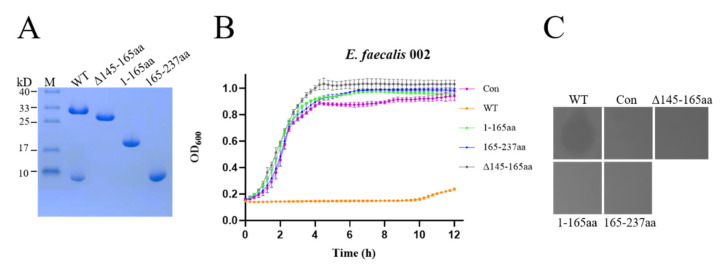
Expression and lytic activity of the EFm1 domains. (**A**) SDS-PAGE results of the EFm1 domain; (**B**) Growth curves of the EFm1 domains; the mean ± s.e.m is shown; (**C**) Plate lysis assay results of the EFm1 domain.

**Figure 4 materials-15-04879-f004:**
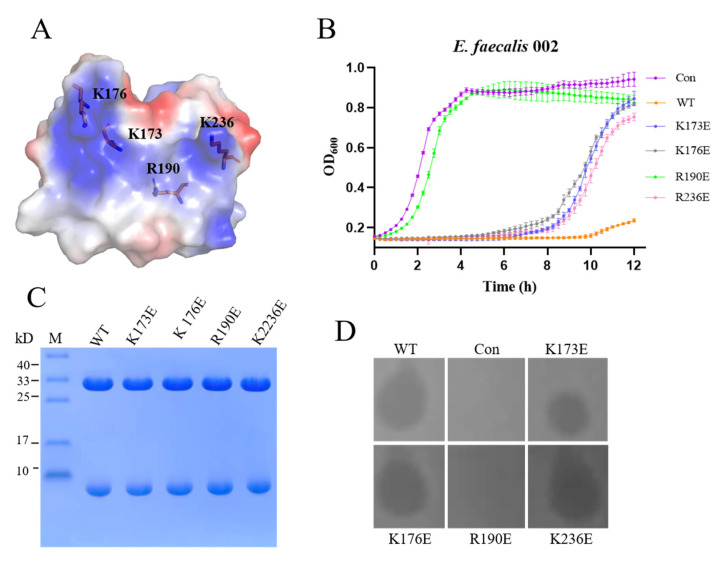
The binding sites of EFm1 for cell wall. (**A**) Residues with a positive charge in the convex region of the monomer; (**B**) Growth curves of the mutants K173E, K176E, R190E and K236E; the mean ± s.e.m is shown; (**C**) SDS-PAGE results of the mutants K173E, K176E, R190E and K236E; (**D**) Plate lysis assay results of the mutants K173E, K176E, R190E and K236E.

**Figure 5 materials-15-04879-f005:**
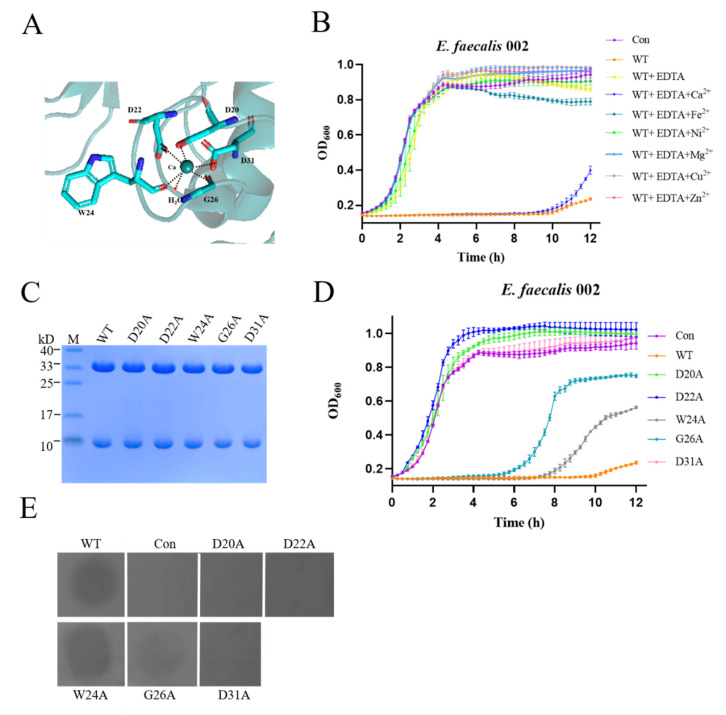
Effect of calcium on the lytic activity of EFm1. (**A**) Residues coordinating with calcium; (**B**) Effects of different metal ions (1 μM) on the lytic activity of EDTA-inactivated EFm1; the mean ± s.e.m is shown; (**C**) SDS-PAGE results of mutants coordinated with calcium; (**D**) Growth curves of mutants coordinated with calcium; the mean ± s.e.m is shown; (**E**) Plate lysis assay results of mutants coordinated with calcium.

**Figure 6 materials-15-04879-f006:**
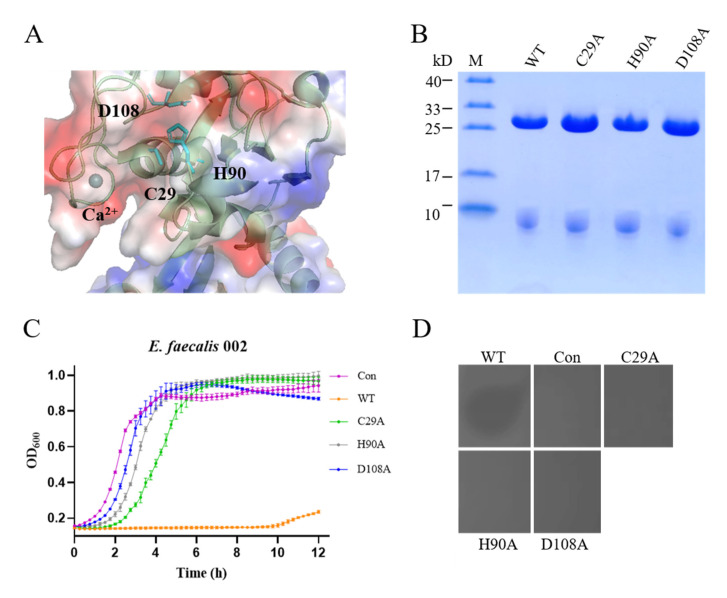
Lytic activity of the amidohydrolase catalytic triad (C29, H90 and D108) in EFm1. (**A**) Structure of the amidohydrolase catalytic triad; (**B**) SDS-PAGE results of mutants with amidohydrolase catalytic triad; (**C**) Growth curves of mutants with amidohydrolase catalytic triad; the mean ± s.e.m is shown; (**D**) Plate lysis assay results of mutants with amidohydrolase catalytic triad.

**Figure 7 materials-15-04879-f007:**
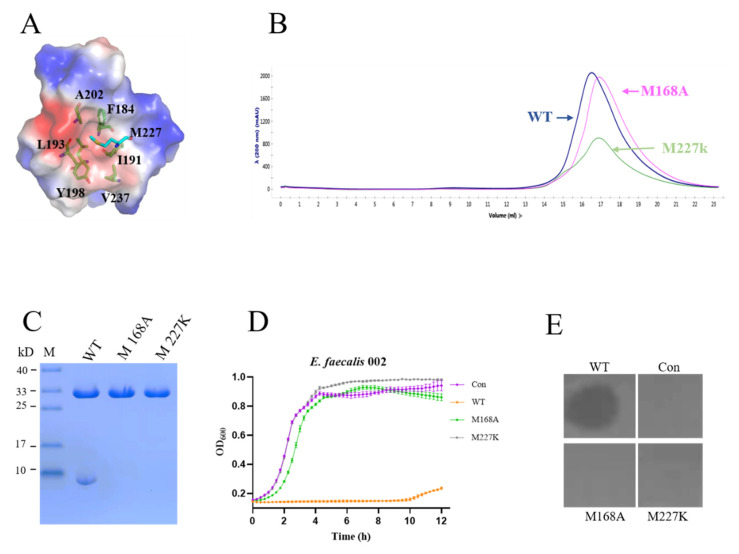
Lytic activity of the EFm1 tetramer. (**A**) Hydrophobic interactions between M227 and the nearby monome; (**B**) Size-exclusion chromatography of WT, M168A and M227K; (**C**) SDS-PAGE results of the mutants M168A and M227K, respectively; (**D**) Growth curves and plate lysis assay results of the mutants M168A and M227K, respectively; the mean ± s.e.m is shown; (**E**) Plate lysis assay results of the mutants M168A and M227K, respectively.

**Figure 8 materials-15-04879-f008:**
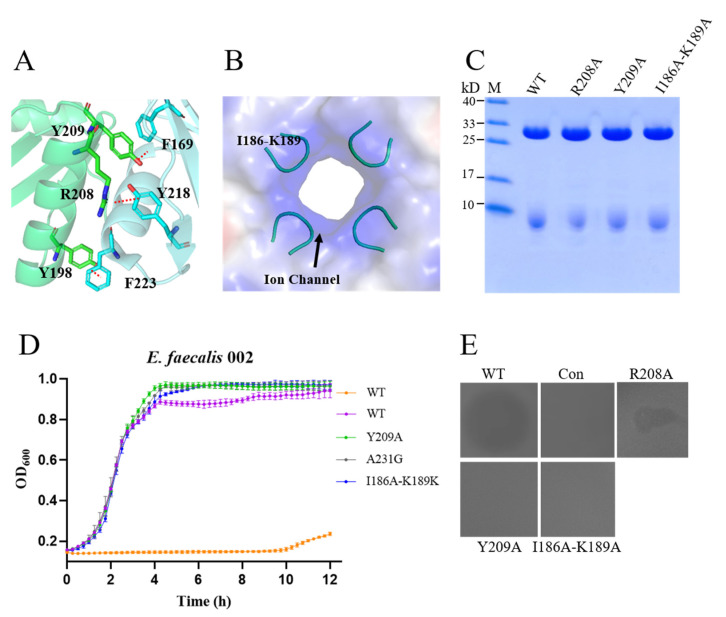
Putative ion channel and interaction between EFm1 monomers. (**A**) Putative ion channel in the CTD of EFm1; (**B**) The interaction between monomers of EFm1; (**C**) SDS-PAGE results of the mutants R208A, Y209A and I186A-K189A, respectively; (**D**) Growth curves of the mutants R208A, Y209A and I186A-K189A, respectively; the mean ± s.e.m is shown; (**E**) Plate lysis assay results of the mutants R208A, Y209A and I186A-K189A, respectively.

**Figure 9 materials-15-04879-f009:**
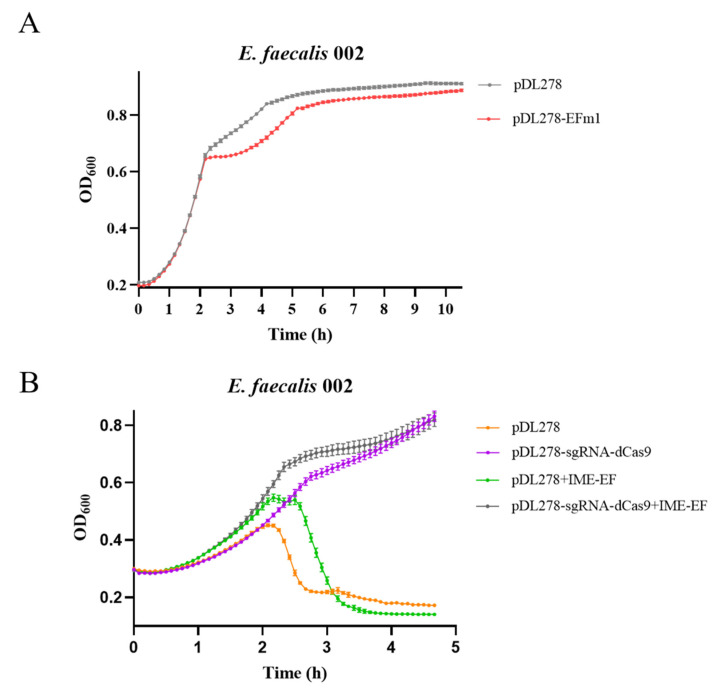
Expression and CRISPRi of EFm1 in 002. (**A**) Growth dynamics of 002 with the plasmid pDL278 or pDL278-*efm1*; the mean ± s.e.m is shown; (**B**) Growth dynamics of 002 with the plasmid pDL278 or pDL278-dCas9-sgRNA followed by infection without or with the phage IME-EF1 (MOI = 0.1); the mean ± s.e.m is shown.

**Table 1 materials-15-04879-t001:** Primers used in this study.

Name	Sequence (5′ to 3′)	Usage
EFm1_1–165 aa_-F	AGACGATTAGCTCGAGCGGCCGCATCGT	pGEX-EFm1_1–165 aa_
EFm1_1–165 aa_-R	GCTCGAGCTAATCGTCTCCTTTAAATTGTCCTAGATT	
EFm1_165–237 aa_-F	CTGGGATCCGATGATATTATGTTCATCTATTACAAACGCAC	pGEX-EFm1_165–237 aa_
EFm1_165–237 aa_-R	ATATCATCGGATCCCAGGGGCCCCTG	
EFm1_△145–165 aa_-F	GCGTTTGTAATAGATGAACATAATATCTGAAGCAGCTTCGTAAGG	pGEX-EFm1_△145–165 aa_
EFm1_△145–165 aa_-R	GATATTATGTTCATCTATTACAAACGCACTAAGCAAGGA	
EFm1_D166Q_-F	GAGACGATCAAATTATGTTCATCTATTACAAACGCACTAA	pGEX-EFm1_D166Q_
EFm1_D166Q_-R	CATAATTTGATCGTCTCCTTTAAATTGTCCTAGATT	
EFm1_C29A_-F	TACGCAAGCGATGGACTTGACAGTAGACGTTATGCA	pGEX-EFm1_C29A_
EFm1_C29A_-R	AGTCCATCGCTTGCGTACCATACCATCCGTC	
EFm1_H90A_-F	ATATGGTGCGACAGGTATCGCAACAGAGGATGG	pGEX-EFm1_H90A_
EFm1_H90A_-R	TACCTGTCGCACCATATTGAGCATAGTATCCTAAGCC	
EFm1_D108A_-F	CAGTGTTGCGCAAAACTGGATTAACCCAAGCC	pGEX-EFm1_D108A_
EFm1_D108A_-R	AGTTTTGCGCAACACTGACAAAGGTTCCGTCAG	
EFm1_D20A_-F	AAAGGCGTGGCGGCTGACGGATGGTATGGTACG	pGEX-EFm1_D20A_
EFm1_D20A_-R	TCAGCCGCCACGCCTTTTCCGACAAGG	
EFm1_D22A_-F	ACGCTGCGGGATGGTATGGTACGCAATGTATG	pGEX-EFm1_D22A_
EFm1_D22A_-R	ATACCATCCCGCAGCGTCCACGCCTTTTCC	
EFm1_W24A_-F	TGACGGAGCGTATGGTACGCAATGTATGGACTTGA	pGEX-EFm1_W24A_
EFm1_W24A_-R	TACCATACGCTCCGTCAGCGTCCACGCC	
EFm1_G26A_-F	TGACGGAGCGACGCAATGTATGGACTTGACAGTAG	pGEX-EFm1_G26A_
EFm1_G26A_-R	ATTGCGTCGCTCCGTCAGCGTCCACGCC	
EFm1_D31A_-F	GTACGCAATGTATGGCCTTGACAGT	pGEX-EFm1_D31A_
EFm1_D31A_-R	ACGTCTACTGTCAAGGCCATACATT	
EFm1_M168A_-F	GAGACGATGATATTGCGTTCATCTATTACAAACGCACTAAGCA	pGEX-EFm1_M168A_
EFm1_M168A_-R	CGCAATATCATCGTCTCCTTTAAATTGTCC	
EFm1_M227A_-F	TTGAAGGCGATGGAAGCAGCTTTACCACAAGT	pGEX-EFm1_M227A_
EFm1_M227A_-R	GCTTCCATCGCCTTCAATCCAAAGTTGTCGTGGT	
EFm1_M227K_-F	GAAGAAGATGGAAGCAGCTTTACCACAAGT	pGEX-EFm1_M227K_
EFm1_M227K_-R	CTGCTTCCATCTTCTTCAATCCAAAGTTGTCGTGGT	
EFm1_K173E_-F	CTATTACGCGCGCACTAAGCAAGGAAGCACT	pGEX-EFm1_K173E_
EFm1_K173E_-R	TAGTGCGCTCGTAATAGATGAACATAATATCATCGTCTCC	
EFm1_K176E_-F	ACGCACTGAGCAAGGAAGCACTGAGCAATGG	pGEX-EFm1_K176E_
EFm1_K176E_-R	TTCCTTGVTVAGTGCGTTTGTAATAGATGAACATAATATC	
EFm1_R190E_-F	GAGGTAAAGAGATCTACTTACCAACAATGACTTACGTAAAC	pGEX-EFm1_R190E_
EFm1_R190E_-R	GTAGATCTCTTTACCTCCAATAACGAACCATTG	
EFm1_K236E_-F	AGTTGAGGTATAGCTCGAGCGGCCGC	pGEX-EFm1_K236E_
EFm1_K236E_-R	CGAGCTATACCTCAACTTGTGGTAAAGCTGCTTCCA	
EFm1_I186A-K189A_-F	ACTGAGCAATGGTTCGTTCGTATCTACTTA	pGEX-EFm1_I186A–K189A_
EFm1_I186A-K189A_-R	TGTTGGTAAGTAGATACGAACGAACCATTG	
EFm1_Y209A_-F	GACCTTATCAAACGAGCTGGTGGAA	pGEX-EFm1_Y209A_
EFm1_Y209A_-R	GTTAGTGTTTCCACCAGCTCGTTTG	
EFm1_R208A_-F	GCATATGGTGGAAACACTAACGTAACGA	pGEX-EFm1_R208A_
EFm1_R208A_-R	GTGTTTCCACCATATGCTTTGATAAGGTCGTTAGCTTCGTT	
pDL278-EFm1-F	GAGCTCGGTACCCGGGGATCCCTATACTTTAACTTGTGGTAAA	pDL278-EFm1
pDL278-EFm1-R	ACCATGATTACGCCAAGCTTAATGGTTAAATTAAACGATGTA	
dCas9-F	GACCATGATTACGCCAAGCTTAATGGACAAGAAGTACAGCATCGGC	pDL278-dCas9
dCas9-R	GAGCTCGGTACCCGGGGATCCTTAGTCGCCGCCCAGCTG	
pDL278-F	ATGACCATGATTACGCCAAGCTTAATGGACAAGAA	Linearized pDL278-dCas9
pDL278-R	TTGTTATCCGCTCACAATTCCACACAACATACGAG	
sgRNA-F	GAATTGTGAGCGGATAACAAGACGTTATGCAACGCTTCTTGTTTTAGAGCTAGAAATAGCAAG	pDL278-dCas9-sgRNA
sgRNA-R	CTTGGCGTAATCATGGTCATCAAAAAAAGCACCGACTCGG	

**Table 2 materials-15-04879-t002:** Data and refinement statistics.

Property	EFm1_D166Q_	CTD
Space group	I 41 2 2	I 41 2 2
Cell constants	46.87 Å 57.26 Å 91.27 Å	115.59 Å 115.59 Å 179.72 Å
a, b, c, α, β, γ	90.00° 96.57° 90.00°	90.00° 96.57° 90.00°
Resolution (Å)	48.46–2.20	49.73–2.00
	48.42–2.20	49.68–2.00
% Data completeness	99.9 (48.46–2.20)	99.0 (49.73–2.00)
(in resolution range)	98.2 (48.42–2.20)	98.2 (49.68–2.00)
R*_merge_*	0.00	0.00
< *I*/σ (*I*) >	2.75 (at 2.20 Å)	1.39 (at 2.00 Å)
Refinement program	REFMAC 5.8.0267, REFMAC 5.8.0267	REFMAC 5.8.0267, REFMAC 5.8.0267
R, R_free_	0.179, 0.215	0.183, 0.216
	0.197, 0.223	0.177, 0.214
R_free_ test set	1295 reflections (5.27%)	1998 reflections (4.91%)
Wilson B-factor (Å)	27.1	21.8
Anisotropy	0.681	0.121
Bulk solvent *k_sol_* (e/Å), *B_sol_* (Å)	0.39, 33.6	0.42, 58.0
L-test for twinning	< |L| > = 0.49, < L^2^ > = 0.32	< |L| > = 0.49, < L^2^ > = 0.32
Estimated twinning fraction	No twinning to report.	No twinning to report.
F*_O_*, F*_C_* correlation	0.94	0.96
Total number of atoms	6953	9609
Average B, all atoms (Å)	24.0	23.0

Statistics for the highest-resolution shell are shown in parentheses.

**Table 3 materials-15-04879-t003:** Summary of the domains and residues identified as being important for the stability and lytic activity of EFm1 in this study.

Structural Characteristics	Mutants	Stability	Lytic Activity
Critical domains	NTD (1–165 aa)	−	−
	CTD (165–237 aa)	−	−
	EFm1 lacking the linker (Δ145–165 aa)	−	−
Putative residues binding to the host cell	K173E	+	+
	K176E	+	+
	R190E	+	−
	K236E	+	+
Residues binding calcium	D20A	+	−
	D22A	+	−
	W24A	+	+
	G26A	+	+
	D31A	+	−
Residues forming a putative catalytic triad	C29A	+	−
	H90A	+	−
	D108A	+	−
Residues producing and assembling the 8 kDa fragment	M168A	−	−
	M227K	−	−
Residues forming the interaction surface	R208A	+	−
	Y209A	+	−
Residues forming the putative ion channel	I186A-K189A	+	−

Note: + and − in the stability column indicate the presence and absence of the 8 kDa protein band, respectively. In the lytic activity column, + indicates that the mutant had lytic activity toward 002, and − indicates that the mutant had no lytic activity toward 002.

## Data Availability

Data are available on request to the authors.

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
