# Peer review of "The Structure and Function of Biomaterial Endolysin EFm1 from *E. faecalis* Phage"

_materials, 2022, doi:10.3390/ma15144879_

Round 1

Reviewer 1 Report

Dear Editor,

Thank you for the opportunity to review the paper entitled "The structure and function of a biomaterial endolysin EFm1 from E. faecalis phage". The paper is well-structured and written in the scientific manner. The obtained results are interesting and useful. Nevertheless, there are a few things that should be added or changed before the further process of publication.

Some of my overall comments for this paper:

-          Abstract part:

line 22: Please put in vivo in italic

-          Introduction part:

The introduction part is generally well-written and concise. The only thing that should be modified relates to the last few sentences in this part.

Please, try to restructure the sentences in lines 74-78 so the word we is avoided.

-          Materials and methods:

Throughout the part Materials and methods the producers of culture medium and/or medium components should be provided.

Table 1 should be supplemented with the appropriate references in which the chosen primers were used.

In the section 2.5., line 162, the parameters for shaking step should be incorporated.

-          Results:

Table 2. is completely unnecessary. According to the presented results only in the case of E. faecalis 945 the protein has no lytic activity. Therefore, the obtained results can also be presented in words.

Please, explain how the fragment length (Figure 1A) is ~8kD? It is obvious that the length of the fragment is approximately 12 kD. Additionaly, for all growth curves in the manuscript the values of standard deviation should be added.

In the section 3.5, lines 279-280, the statement related to the effects of calcium in regulation of protease activity should be extensively explained.

-          Discussion:

Line 388, please, correct it lytic ability to its lytic ability.

-          Conslusions:

Line 440, please, correct we couldn not clearly explains to we could not clearly explain

Line 443, correct the following: Not only are further investigation of EFm1 important…. to Not only the further investigation of EFm1 is important…

-          References:

The choice of the references are good and up to dated.               

Author Response

Reviewer 1

Comment: Abstract part: line 22: Please put in vivo in italic.

Response: The in vivo was put in italic in line 24 of R1.

Comment: Modified relates to the last few sentences in introduction at lines 74-78.

Response: We have changed as below: In this study, a strain E. faecalis 945 resisted to EFm1 is obtained by a deeply broaden the cleavage spectrum of EFm1; lytic mechanism of EFm1 is identified by determining the structures of EFm1D166Q and EFm1166-237aa, analyzing their lytic activity and verifying activity of efm1 in 002 and IME-EF1.

Comment: Throughout the part Materials and methods the producers of culture medium and/or medium components should be provided.

Response: The producers of culture medium and/or medium components were provided in the Materials and Methods of R1.

Comment: Table 1 should be supplemented with the appropriate references in which the chosen primers were used.

Response: The appropriate reference (ref. 17) in which the chosen primers were used was supplemented in Plasmid Construction of R1.

Comment: In the section 2.5., line 162, the parameters for shaking step should be incorporated.

Response: The parameters for shaking step was incorporated in line 169 of R1.

Comments: Table 2. is completely unnecessary. According to the presented results only in the case of E. faecalis 945 the protein has no lytic activity. Therefore, the obtained results can also be presented in words.

Response: The Table 2 was deleted and the obtained results were presented in words in lines 205-207 of R1.

Comment: Please, explain how the fragment length (Figure 1A) is ~8kD? It is obvious that the length of the fragment is approximately 12 kD. Additionaly, for all growth curves in the manuscript the values of standard deviation should be added.

Response: As you mentioned, the fragment length was approximately 12 kD by SDS-PAGE analysis. The fragment length was further identified by Mass spectrometric analysis, confirming the fragment length was ~8 kD. The possible reason is that the protein Marker is inaccurate.

The values of standard deviation for all growth curves in R1 were added.

Comment: In the section 3.5, lines 279-280, the statement related to the effects of calcium in regulation of protease activity should be extensively explained.

Response: The statement related to the effects of calcium in regulation of protease activity was extensively explained in lines 295-300 of R1. The revised statement as below: Protease easily recruit metal ions to the active center in calcium rich environment of the gram-positive cell; the stable states of protease are regulated by the number of calcium binding sites [24]. When all calcium binding sites are occupied, protease can fold to a stable, active conformation. Therefore, calcium usually serves as a switch to regulate the activity of proteases. So, atoms coordinating with calcium could affect the activity of proteases by changing the spatial position of calcium, such as subtilisin or alpha-lytic protease.

Comment: Line 388, please, correct it lytic ability to its lytic ability.

Response: it lytic ability” was corrected to “its lytic ability” in line 416 of R1.

Comment: Line 440, please, correct we couldn not clearly explains to we could not clearly explain.

Response: Thanks. We have corrected.

Comment: Line 443, correct the following: Not only are further investigation of EFm1 important…. to Not only the further investigation of EFm1 is important…

Response: Thanks. We have corrected.

Comment: The choice of the references needs to dated.

Response: Thanks. We have changed some references to new references, as ref. [5,6] of R1.

Reviewer 2 Report

The manuscript entitled "The structure and function of a biomaterial endolysin EFm1 from E. faecalis phage" represents detailed research for characterizing the endolysin EFm1 of a bacteriophage against E. faecalis. The detailed characterization of phage endolysins via crystallography and other methodologies is very useful and relevant to the field. This detailed characterization rarely takes place in phage endolysins so compared to other studies represents lots of new information. The authors seem experts in the field of crystallography. The manuscript seems that has already been reviewed in detail in the past and the methodologies is very detailed and completed. Due to the scope of the journal I believe the authors present a complete story regarding this protein. Authors discuss domains the newly found domains of the protein and hypothesize about possible efficacy application in vitro 

The manuscript presents some great experimental details and results which end up characterizing 90% of the endolysin. 

I am a little concerned regarding the term used "biomaterial". I was expecting to read about an artificial biomaterial with possible augmented endolysin in it but in the end, it was a thorough characterization of an endolysin of interest. This has to be more clear in the last sentence of the abstract. 

I only have a few minor comments.

Figures

Try to enhance the contrast in figures whereas petri dish photographs are taken. The difference between control and mutants cannot be seen easily. 

Minor spellings in lines 440, and 431 (it is).

Please write in materials and methods the accession number of the bacteriophage genome in GenBank. 

Author Response

Reviewer 2

Comment:  I am a little concerned regarding the term used "biomaterial". I was expecting to read about an artificial biomaterial with possible augmented endolysin in it but in the end, it was a thorough characterization of an endolysin of interest. This has to be more clear in the last sentence of the abstract.

Response: Thanks. We have added as below: "Therefore, EFm1 may be an alternative biomaterial to prevent and treat diseases caused by E. faecalis" in lines 16-17 of R1. " These findings provide valuable insights into the molecule mechanism of a potential functional biomaterial for treatment of the disease caused by the opportunistic pathogen E. faecalis." in lines 25-27 of R1.

Comment: Try to enhance the contrast in figures whereas petri dish photographs are taken. The difference between control and mutants cannot be seen easily.

Response: Thanks. The contrast in figures of petri dish photographs were enhanced in R1 and all revised figures were uploaded.

Comment: Minor spellings in lines 440, and 431 (it is).

Response: Thanks. We have corrected.

Comment: Please write in materials and methods the accession number of the bacteriophage genome in GenBank.

Response: Thanks. The accession number of the bacteriophage genome in GenBank in the Phage and Strains were added in R1. 
